# Robust optimal control of interacting multi-qubit systems for quantum sensing

Nguyen H. Le,[1] Max Cykiert,[1] and Eran Ginossar[1]

[1]*Advanced Technology Institute and Department of Physics,*
*University of Surrey, Guildford GU2 7XH, United Kingdom*

Realising high fidelity entangled states in controlled quantum many-body systems is challenging due to experimental uncertainty in a large number of physical quantities. We develop a robust optimal control method for achieving this goal in interacting multi-qubit systems despite significant uncertainty in multiple parameters. We demonstrate its effectiveness in the generation of the Greenberger-Horne-Zeilinger state on a star graph of capacitively coupled transmons, and discuss its crucial role for achieving the Heisenberg limit of precision in quantum sensing.

## I. INTRODUCTION

Recent advances in quantum technology have enabled the building of multi-qubit quantum devices across many platforms such as cold atoms, trapped ion and superconducting qubits. Among its goal is the creation of multi-qubit entangled states, such as the Schrodinger's cat, for understanding macroscopic quantum physics [1, 2] and for applications in quantum sensing [3–5]. The most recent remarkable results are the realisations of a 20 qubit Greenberger-Horne-Zeilinger (GHZ) state in an array of Rydberg atoms [1] and superconducting qubits [2], but the fidelity in these experiments is lower than 60% due to the complexity of controlling the many-body dynamics. In a large device it is impossible to precisely characterise the physical parameters of every qubits, resulting in some uncertainty with regards to the model of the system. These parameters also change from one experiment to the next due to diffusive processes in the materials and loss of calibration in the external electronic equipment. This uncertainty leads to a significant drop in the fidelity of the state preparation process, an effect that becomes rapidly worse with an increasing number of the uncertain parameters. Therefore, it is important to have a theoretical method for preparing entangled states with high fidelity and robustness against experimental uncertainty in multi-qubit systems.

In this paper we develop a robust optimal control algorithm for interacting multiqubit systems. It can be used to find optimal pulses for creating entangled sates with high fidelity despite uncertainty in multiple parameters in the Hamiltonian. As a concrete example, we study the generation of the GHZ state on a system of transmons connected by simple capacitive couplers. All the coupling strengths between the transmons may vary independently in an uncertain interval up to 5% around the mean. The effect of the excitation to the higher levels of the transmons, also known as the population leakage, is considered. When this GHZ state is used for quantum sensing, we show how parameter uncertainty is detrimental to the precision of the measurement result, and how robust optimal control leads to a big improvement and is crucial for achieving the Heisenberg bound. In particular, for a cluster of 8 transmons with 5% uncertainty in all the coupling strengths, the precision increases by up to a factor of 5 after robust optimal control is applied.

Computing the fidelity of a quantum evolution of many-body systems is computationally expensive due to the exponential growth of the size of the wave-function with the number of particles/qubits. Optimising the fidelity introduces another layer of complexity as it requires the fidelity to be calculated for many iterations. Adding robustness against uncertainty in the Hamiltonian's parameter is even more challenging because one needs to optimise for a number of Hamiltonian configurations that is exponential in the number of uncertain parameters. We use a combination of techniques to make many-body robust optimal control as efficient as possible: 1) the Krylov-subspace method [6] for computing the unitary exponential of a sparse Hamiltonian, 2) the observation that the worst-case fidelity lies at the extreme points of the convex region of the uncertain parameters, thus one needs only compute the fidelity at these points, and 3) for qubit clusters with high symmetry many extreme points give the same fidelity, reducing significantly the number of distinct fidelities one must compute. For the generation of the GHZ state we consider a star graph of coupled transmons. The simple geometry of a star graph helps keeping the number of control channels at a minimum, and its high symmetry reduces the number of the distinct fidelities from exponential to *only linear* in the number of uncertain parameters, resulting in a huge speed up of the optimisation.

## II. ROBUST OPTIMAL CONTROL OF INTERACTING QUBITS

We consider an interacting system of $N$ qubits, or qudits, where the bare Hamiltonian of the j-th qubit is $Q_j$, the coupling operators between the qubits are $V_{jk}$ and the coupling strength $J_{jk}$. A set of qubits, denoted by $\mathcal{L}$, are controlled with two-quadrature driving fields, $\Omega_j^x$ and $\Omega_j^y$, through the field-qubit coupling operators, $S_j^x$ and $S_j^y$. The system's Hamiltonian in a frame rotating with the frequencies of the drives is then

$$H(t) = \sum_{j=1}^{N} Q_j + \sum_{j,k=1}^{N} J_{jk} V_{jk} + \sum_{j \in \mathcal{L}} \left( \Omega_j^x(t) S_j^x + \Omega_j^y(t) S_j^y \right).$$

(1)

For the ideal case where all the qubits are two-level systems coupled with each other through the flip-flop interaction and coupled with the drive through the dipole interaction, $Q_j = -\Delta_j \sigma_j^z/2$ where $\Delta_j$ is the $j$-th qubit's detuning, $V_{jk} = \sigma_j^+ \sigma_k^- + h.c.$, and $S_j^{x,y} = \sigma_j^{x,y}$. Here $\sigma^{x,y,z}$ are the Pauli matrices and $\sigma^\pm = \sigma^x \pm \sigma^y$. $\Delta_j$ can be chosen to be zero for the undriven qubits. For superconducting transmons the multi-level structure has to be considered and one has to use the multi-level generalisation of the above operators (see more details below).

The coupling terms $V_{jk}$ are typically much smaller than the energy separation of the qubits. Thus, the ground state of the system when not driven is $|\psi_g\rangle = \otimes_{j=1}^{N} |0\rangle_j$ where $|0\rangle_j$ is the ground state of $Q_j$. Now one can design a pulse with duration $T$ for realising a target multi-qubit entangled state, $|\psi_{\text{tg}}\rangle$. In reality the final state at the end of the pulse, $|\psi\rangle \equiv U(T)|\psi_g\rangle$ with $U(T)$ the unitary evolution operator, is not exactly $|\psi_{\text{tg}}\rangle$ and the fidelity of the process, defined as the overlap

$$F = |\langle \psi_{\text{tg}} | U(T) | \psi_g \rangle |^2, \qquad (2)$$

is less than one. The goal of quantum optimal control is to find the pulse shape of $\Omega_j^{x,y}(t)$ that maximises $F$.

The Hamiltonian, $H(t)$, depends on many physical parameters such as the transition frequencies of the qubits, the coupling strengths between the qubits, and the magnitude of the field-qubit couplings. All of these parameters have experimental uncertainty due to limited calibration precision and slow drift due to diffusion processes in the material. When there are a large number of uncertain parameters the final state varies sharply in the uncertain region and so does the fidelity. Robust optimal control is a technique for finding optimal shapes such that the fidelity is high regardless of the actual values of the physical parameters in the uncertain region. The final state is thus robust against the variation in these parameters. In this paper we assume that the only uncertain parameters are the qubit-qubit coupling strengths, $J_{jk}$, as these are typically the hardest to measure and calibrate in most physical realisations of qubits. The formulation described below can be easily applied to an arbitrary set of uncertain parameters.

We divide the drive's duration $T$ into $m$ equal intervals with $t_0 = 0$ and $t_m = T$, and we assume a piecewise control pattern where the driving amplitudes $\Omega_j^{x,y}(t)$ are constant in each interval. Denote by $\boldsymbol{c}$ the control vector of all the $2mN_\mathcal{L}$ control variables in the set $\{\Omega_{jn}^\mu : \mu = x, y; j \in \mathcal{L}; 1 \le n \le m\}$ where $N_\mathcal{L}$ is the number of qubits in the driven subset, and denote by $\boldsymbol{v}$ the set of all the uncertain physical parameters in the Hamiltonian, which in our case includes $J_{jk} : 1 \le j, k \le N$. Each $J_{jk}$ is allowed to vary *independently* in the interval $[\bar{J} - \Delta J/2, \bar{J} + \Delta J/2]$, and thus the vector $\boldsymbol{v}$ takes values in a hypercube whose volume is $|\Delta J|^{n_v}$ where $n_v$ is the number of uncertain parameters. Obviously the unitary evolution and hence the fidelity in Eq. (2) is a multi-variable function of $\boldsymbol{c}$ and $\boldsymbol{v}$, and robust optimal control can be defined as a max-min optimisation problem where we find the control that maximises the minimum fidelity over $\boldsymbol{v}$, referred to as the worst-case fidelity:

$$\text{Find } \mathcal{F}_{\max} = \max_{\boldsymbol{c}} \mathcal{F}(\boldsymbol{c}), \quad \mathcal{F}(\boldsymbol{c}) = \min_{v \in \mathcal{V}} F(\boldsymbol{c}, \boldsymbol{v}), \qquad (3)$$

where $\mathcal{V}$ is the hypercube containing all the possible values of $\boldsymbol{v}$.

In numerical computation one chooses a set of sampling points $\boldsymbol{v}_i$ in $\mathcal{V}$, and find the minimum fidelity in this set. The number of sampling points is exponential in the number of the uncertain parameters: If one chooses $n_s$ equal spacing points in the uncertain interval for each parameter, then the total number of sampling points in $\mathcal{V}$ is $n_s^{n_v}$. For our specific problem we find that when the uncertainties are all smaller than 5% the fidelity function $F(\boldsymbol{c}, \boldsymbol{v})$ can be made to be concave in $\boldsymbol{v}$, i.e., its maximum over $\boldsymbol{v}$ is in the interior, and its minimum always lies at one of the extreme points of $\mathcal{V}$, i.e., one of the corners of the hypercube (we describe how to do this numerically in Sec. II B). Therefore, we can redefine the minimum fidelity over $\mathcal{V}$ as

$$\mathcal{F}(\boldsymbol{c}) \equiv \min_{\boldsymbol{v}_i \in \mathcal{X}} F(\boldsymbol{c}, \boldsymbol{v}_i), \qquad (4)$$

where $\mathcal{X}$ is the set of the extreme points of $\mathcal{V}$. There are $2^{n_v}$ extreme points in the hypercube, which is still exponential but this is in general the smallest number of fidelities one must compute in robust optimal control. If there is symmetry in the system and the target state, many extreme points give the same fidelity. For the example of the highly symmetric GHZ state on a star graph described below the number of distinct fidelities is only linear in $n_v$.

## A. Calculating the fidelity and its gradient

Since piecewise pulses are used, the Hamiltonian is constant in each time step from $t_{n-1}$ to $t_n$ and the unitary evolution is $U_n = e^{-iH_n \Delta t}$, where $\Delta t = T/m$ and $H_n$ is the Hamiltonian during the $n$-th time step. For a multi-qubit system the size of the Hamiltonian matrix increases exponentially with the number of qubits and computing the matrix exponentiation is very costly in both memory and time. However, one needs only the product of the unitary matrix and a state vector, $|\psi_n\rangle = U_n |\psi_{n-1}\rangle$, and one can use the efficient Krylov subspace algorithm to compute it directly (without computing the matrix exponential). This method is capable of handling very large sparse matrices [6], and in our test it works for systems with up to around 20 qubits.

For an efficient calculation of the fidelity and its gradients we *compute and store* all the forward and backward propagating states [7], defined by

$$|\psi_n^f\rangle = U_n U_{n-1} \ldots U_1 |\psi_0\rangle,$$
$$\langle \psi_{n+1}^b | = \langle \psi_{tg} | U_M U_{M-1} \ldots U_{n+1},$$

using the recursive relations $|\psi_n^f\rangle = U_n|\psi_{n-1}^f\rangle$ and $\langle\psi_{n+1}^b| = \langle\psi_{n+2}^b|U_{n+1}$. This costs $2m$ Krylov multiplications and is the most expensive part of the calculation. The fidelity is then simply $F(\boldsymbol{c},\boldsymbol{v}) = |\langle\psi_{\text{tg}}|\psi_m^f\rangle|^2$.

For calculating the gradients used in the optimisation algorithms we note that

$$H_n = H_0 + \sum_{\mu=x,y}\sum_{j\in\mathcal{L}}\Omega_{jn}^\mu S_j^\mu, \qquad (5)$$

where $H_0 = \sum_{j=1}^N Q_j + \sum_{j,k=1}^N J_{jk}V_{jk}$ is independent of the control variables. One can show that the derivatives of $U_n \equiv e^{-iH_n\Delta t}$ with respect to the control variables, $\Omega_{jn}^\mu$, are [7]

$$\frac{\partial U_n}{\partial\Omega_{jn}^\mu} = \left\{-i\Delta t S_{jn}^\mu + \frac{\Delta t^2}{2}\left[H_n, S_j^\mu\right]\right\}U_n + O(\Delta t^3), \qquad (6)$$

where $\left[H_n, S_j^\mu\right]$ is a commutator. It follows that the derivatives of $\langle\psi_{\text{tg}}|\psi_m^f\rangle \equiv \langle\psi_{n+1}^b|U_n|\psi_{n-1}^f\rangle$ are

$$\frac{\partial\langle\psi_{n+1}^b|U_n|\psi_{n-1}^f\rangle}{\partial\Omega_{jn}^\mu} = \langle\psi_{n+1}^b|\frac{\partial U_n}{\partial\Omega_{jn}^\mu}|\psi_{n-1}^f\rangle$$

$$\approx \langle\psi_{n+1}^b|\left\{-i\Delta t S_j^\mu + \frac{\Delta t^2}{2}\left[H_n, S_j^\mu\right]\right\}|\psi_n^f\rangle, \qquad (7)$$

which is obtained by matrix-vector multiplications as $\langle\psi_{n+1}^b|$ and $|\psi_n^f\rangle$ are already computed and stored previously. From this it is straightforward to calculate the gradients of the fidelity in Eq. (2).

## B. Optimisation

We first optimise the fidelity using gradient ascent at the central point of the hypercube $\mathcal{V}$ to a value very close to one, $1 - 10^{-7}$ in our calculation, so that the fidelity is a concave function with a maximum at the centre and minimum at one of the extreme points of $\mathcal{V}$. We then maximise the worst-case fidelity, $\mathcal{F}(\boldsymbol{c})$, defined in Eq. 4. The first approach is based on the sequential convex programming [8]. We start with a random sample of different initial guesses for the control vector $\boldsymbol{c}$. Next, we choose a random positive value vector, $\boldsymbol{u}_0$, for the upper limit of change in the control vector $\boldsymbol{c}$ (trust region). Then a step $|\delta\boldsymbol{c}| < \boldsymbol{u}_0$ is found to maximise $\min_i \nabla_{\boldsymbol{c}}F(\boldsymbol{c},\boldsymbol{v}_i).\delta\boldsymbol{c}$ where $\boldsymbol{v}_i$ is an extreme point of $\mathcal{V}$, i.e., maximise the minimum fist order increment. This ensures that all the fidelities at the extreme points are increased. The above problem can be solved by sequential convex programming (SCP) [8, 9]. We used the YALMIP toolbox and SPDT3 package in Matlab for this purpose. If a step can be found such that $\min_i \nabla_{\boldsymbol{c}}F(\boldsymbol{c},\boldsymbol{v}_i).\delta\boldsymbol{c}$ is positive then we increase the trust region $\boldsymbol{u}_0$ by 1.15, otherwise we decrease it by 2. We choose these factors as they give the fastest convergence in our numerical tests. The procedure is repeated until either the maximum iteration is reached or the trust region drops below a small tolerance.

The second approach is to simply maximise the average fidelity,

$$\bar{\mathcal{F}}(\boldsymbol{c}) = \sum_{i=1}^{n_\mathcal{X}} F(\boldsymbol{c},\boldsymbol{v}_i)/n_\mathcal{X}, \qquad (8)$$

using a quasi-Newton method. Here $n_\mathcal{X}$ is the number of extreme points. Obviously this does not guarantee that the worst-case fidelity is increased, as it is possible for the average to go up while the smallest does not. However, we find in our calculation that the worst-case fidelity is always improved substantially when we maximise the average fidelity. We optimise $\bar{\mathcal{F}}(\boldsymbol{c})$ using the interior-point method implemented in Matlab's fmincon function, where the Hessian is computed from the exact gradients using the BFGS approximation. In our numerical tests the first algorithm is more sensitive on the initial guesses of the control variables. For the multiqubit systems the computation is very expensive and hence it is not practical to run the optimisation with too many initial guesses. We find that for the same running time the second algorithm gives higher fidelities.

At the end of the optimisation we verify the concavity of $F(\boldsymbol{c}_{\text{final}},\boldsymbol{v})$ by calculating the fidelity at randomly generated points in the hypercube $\mathcal{V}$ to confirm that the worst-case fidelity indeed lies at one of the extreme points.

## III. GHZ STATE OF CAPACITIVELY COUPLED TRANSMONS

We now demonstrate the effectiveness of robust optimal control for creating the GHZ state,

$$|\text{GHZ}\rangle = \frac{1}{\sqrt{2}}\left(|0\rangle^{\otimes N} + |1\rangle^{\otimes N}\right), \qquad (9)$$

with high and robust fidelity on a network of interacting qubits. One has the freedom in choosing the geometry of the network. We consider a star graph of identical qubits coupled by the flip-flop interaction where *only the central qubit* is driven, as shown in Fig. 1. This geometry helps minimise the number of control channels. Each $J_{jk}$ takes value in the interval $[J_<, J_>] \equiv [\bar{J} - \Delta J/2, \bar{J} + \Delta J/2]$ where $\bar{J}$ is the mean. At the extreme points of the hypercube $J_{jk}$ is equal to either the lower limit, $J_<$, or the upper limit, $J_>$. From the symmetry of the GHZ state and the star graph one sees that interchanging any two coupling strengths in the graph does not change the fidelity. Therefore, two extreme points of the hypercube give distinct fidelities only if they have a different number of $J_>$. The extreme points $v_i$ can hence be divided into distinct groups with $0, 1, \ldots, n_J$ values of $J_>$, where $n_J \equiv N-1$ is the number of couplings in the graph and is also the length of the vectors $v_i$. All extreme points in the

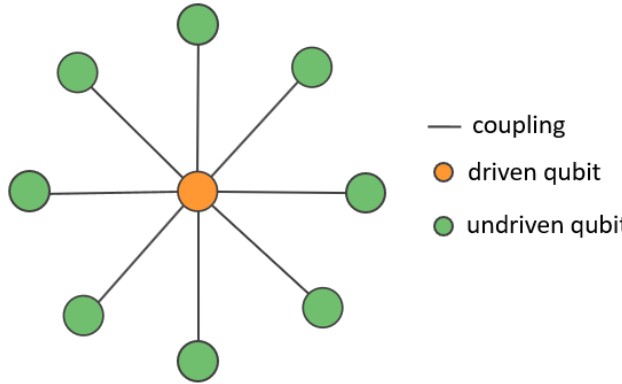

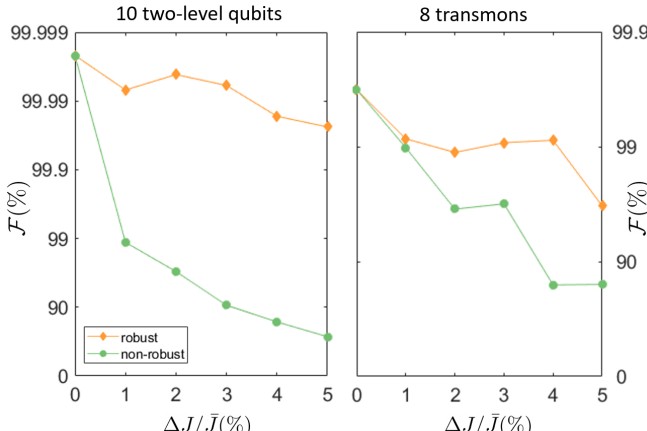

FIG. 1. Controled systems of interacting qubits. A star graph with undriven qubits on the boundary coupled to a driven qubit in the center. The qubits on the boundary are coupled with the central qubit by the flip-flop interaction (see text).

FIG. 2. Optimal fidelity. The worst-case fidelity versus uncertainty levels for a system of 10 two-level qubits (left panel) and 8 multilevel transmons (right panel). Results obtained with robust optimal control are marked by solid diamonds, and non-robust optimal control by solid circles.

same group give the same fidelity. Therefore, the number of distinct extreme points is only $n_J + 1 \equiv N$, which is *linear* in the number of uncertain parameters instead of exponential, a drastic reduction in computational cost. Specifically, this utilisation of symmetry reduces the computation time by a huge factor of $2^{n_J}/(n_J + 1)$ which is around 100 for $n_J = 10$ and 50000 for $n_J = 20$, for example.

Assuming that the central qubit is driven on resonance and each qubit is a two-level system (TLS), the Hamiltonian of the star graph in the rotating wave approximation (RWA) is

$$H_{\text{TLS}}(t) = \sum_{j \in \mathcal{B}} J_j \left( \sigma_j^+ \sigma_d^- + \sigma_j^- \sigma_d^+ \right) + \Omega_d^x(t)\sigma_d^x + \Omega_d^y(t)\sigma_d^y,$$
(10)

where $\mathcal{B}$ is the set of the undriven qubits on the boundary and $d$ indicates the driven qubit at the centre. A good physical realisation of this model is a system of transmons coupled with fixed capacitive coupling [10]. However, the transmon is a multi-level system, and the third level can be populated during the application of the pulses [9, 11]. Thus, it is important to include the higher levels in the model for an accurate calculation of the time evolution. The three-level Hamiltonian for a star graph of transmons in the RWA is

$$H_{\text{tm}}(t) = \sum_j Q_j + \sum_{j \in \mathcal{B}} J_j \left( S_j^+ S_c^- + S_j^- S_c^+ \right)$$
$$+ \Omega_c^x(t) S_c^x + \Omega_c^y(t) S_c^y,$$
(11)

where

$$Q_j = \sum_{k=0}^{2} \left( \omega_j^{(k)} - k\omega_j^{(01)} \right) |k\rangle_j \langle k|_j$$
(12)

is the bare three-level Hamiltonian with $\omega_j^k$ the energy of the eigenstate $|k\rangle_j$ of the $j-$th transmon and $\omega_j^{(10)} = \omega_j^{(1)} - \omega_j^{(0)}$. If one chooses $\omega_j^{(0)} = 0$ then $Q_j = \delta_j |2\rangle_j \langle 2|_j$ where $\delta_j = \omega_j^{(2)} - 2\omega_j^{(1)}$ is the anharmonicity determining how well separated the 1-2 transition is from the 0-1 qubit transition [11, 12]. The coupling operators are

$$S_j^+ = \frac{1}{n_j^{(10)}} \sum_{k=0}^{2} n_j^{(k+1,k)} |k+1\rangle_j \langle k|_j,$$

$$S_j^- = \frac{1}{n_j^{(01)}} \sum_{k=0}^{2} n_j^{(k,k+1)} |k\rangle_j \langle k+1|_j,$$

$$S_j^{x,y} = S_j^+ \pm S_j^-.$$
(13)

The physical parameters of the transmons in our calculation is shown in Table I. The matrix elements $n^{(k,k+1)} \equiv \langle k|n|k+1 \rangle$ of the charge operator $n$ can be calculated from the ratio of the Josephson energy over the charging energy, $E_J/E_C$, and the gate charge, $n_g$ [12].

| Parameters | Symbols | Values |
|---|---|---|
| Qubit transition frequency | $\omega^{(10)}/2\pi$ | 5 GHz |
| Anharmonicity | $\delta/2\pi$ | 300 MHz |
| Mean coupling strength | $\bar{J}$ | 30 MHz |
| Pulse duration | $T$ | 400 ns |
| Transmon energy ratio | $E_J/E_C$ | 50 |
| Transmon gate charge | $n_g$ | 0.25 |

TABLE I. Physical parameters of the transmons and the values used in our calculation.

In Fig. 2 we show the worst-case fidelities obtained with robust optimal control at various levels of uncertainty for the star graph of 10 two-level qubits and that of 8 transmons (the two-level qubit is obtained from the

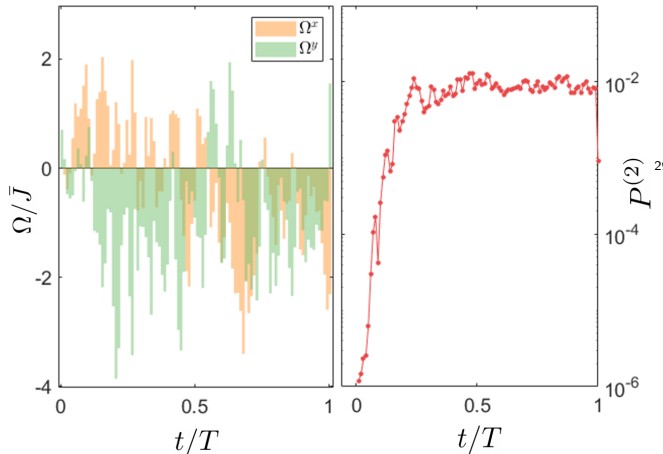

FIG. 3. Pulse shapes. Left panel: Optimal pulse shapes for achieving the fidelity for creating the 8-transmon GHZ state at 5% uncertainty in Fig 2. Right panel: Leakage to the third levels of the transmons during the dynamics, which is the main reason limiting the fidelity.

transmon by simply neglecting the third and higher levels). In this calculation the pulse duration is divided into 100 time bins each with a duration of 4ns. The optimisation requires around 2000 iterations and takes a few hours on a 8-core CPU. For comparison we also show the corresponding fidelities obtained with non-robust optimal control: In this case the optimal pulse is found for the ideal scenario where the uncertainties are neglected, i.e., $J_j = \bar{J}$ for all $j$, then this same pulse is used for calculating the minimum fidelity in the uncertain region at various uncertainty levels.

For both two-level qubits and transmons the non-robust fidelity drops sharply even for small uncertainty, which is due to the combined effect of many uncertain parameters in the quantum dynamics. At $\Delta J/\bar{J} = 5\%$ the robust optimal control improves the fidelity by three nines for two-level qubits and one nine for transmons. It is harder to boost the fidelity for transmons due to the excitation to the third level (population leakage). As a result the driving cannot be too strong, a constraint that reduces the ability of the optimiser for correcting the quantum dynamics under uncertainty.

The optimal pulse shape for the case of transmons at 5% uncertainty is shown in Fig. 3. Although the pulses appear to be rapidly varying, their Fourier transforms have a bandwidth (FWHM) of only 100 MHz. This bandwidth and the 4ns time bin are both within the capability of modern microwave generators such as the Operator X [13]. It might also be possible to obtain smoother pulses if ones implement filtering in the optimisation algorithm [14]. The right panel shows the leakage to the third level, defined as the total population of the third level of all transmons in the star graph, $P^{(2)} = \sum_{j=1}^{N} P_j^{(2)}$ where $P_j^{(2)}$ is the population of $|2\rangle_j$. The remaining leakage at the end of the pulse is the main reason for the rela-

tively low fidelity of tranmons compared with two-level systems.

## IV. PRECISION ENHANCEMENT FOR QUANTUM SENSING

Multi-qubit entangled states are key to quantum sensing, allowing the measurement precision to be improved above the shot noise limit. As an example, suppose $N$ two-level qubits in a GHZ state are subjected to an external magnetic field which induces a phase shift in the upper level, $|1\rangle \rightarrow e^{i\theta} |1\rangle$, then the GHZ state is transformed to $\left(|0\rangle^{\otimes N} + e^{iN\theta} |1\rangle^{\otimes N}\right)/\sqrt{2}$. One can now measure the phase shift by measuring the operator $M = \sigma_x^{\otimes N}$, which produces the following expectation value and variance

$$\langle M \rangle = \cos(N\theta),$$
$$\Delta M^2 = \langle M^2 \rangle - \langle M \rangle^2 = 1 - \cos^2(N\theta). \quad (14)$$

Thus, $\theta$ can be estimated from $\langle M \rangle$. More importantly, the variance in $\theta$, given by the error propagation formula, is

$$\Delta\theta^2 = \Delta M^2 \left/ \left(\frac{\partial \langle M \rangle}{\partial \theta}\right)^2 \right. = \frac{1}{N^2}, \quad (15)$$

which scales as $1/N^2$. This scaling is called the Heisenberg limit and is proved to be the best precision that can be achieved in principle [4]. It is a huge improvement over the shot-noise limit, $1/N$, typical in classical sensing.

A high precision realisation of the entangled state is thus crucial for achieving the $1/N^2$ Heisenberg bound. If, for instance, the state is a product state $\left[(|0\rangle + |1\rangle)/\sqrt{2}\right]^{\otimes N}$ which transforms under the external field to $\left[(|0\rangle + e^{i\theta} |1\rangle)/\sqrt{2}\right]^{\otimes N}$, one can verify easily that $\langle M \rangle = (\cos\theta)^N, \Delta M^2 = 1 - (\cos\theta)^{2N}$ and $\Delta\theta^2 \sim 1/N$ for small $\theta$, which is no better than the classical shot-noise limit. We expect that the precision of quantum sensing decreases significantly if the fidelity of the entangled state generation process is low. Therefore, robust optimal control is very useful for ensuring the advantage of quantum sensing when there are significant parameter uncertainties in the multi-qubit probe. To demonstrate this point for the star graph of transmons, we consider the situation where an external field induces a phase shift $\theta$ in the $|1\rangle$ state of every transmons, transforming the system's quantum state to $|\theta\rangle = U^{\otimes N} |\psi\rangle$, where $U \equiv e^{i\theta} |1\rangle \langle 1| + \sum_{k \neq 1} |k\rangle \langle k|$ is the diagonal unitary matrix that produces the phase shift, and $|\psi\rangle$ is the state obtained at the end of the optimisation in Fig. 2. We take the state for 8 transmons at 5% uncertainty where the fidelity of the GHZ-state-preparation process is 92% for the robust case and only 77% for the non-robust case. The result for the expectation value, $\langle M \rangle$, and the phase shift variance, $\Delta\theta^2$, is shown in Fig. 4. While robust

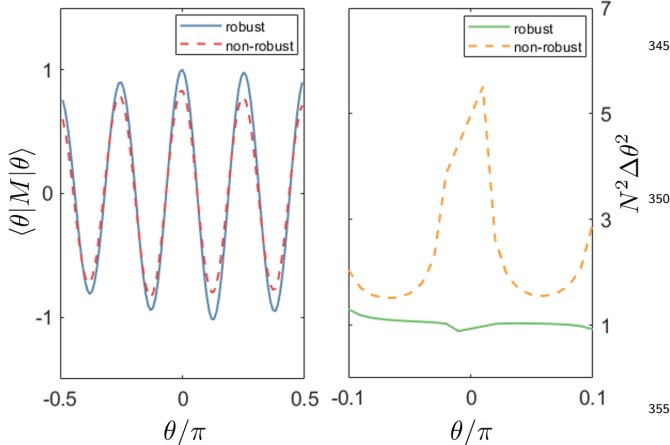

FIG. 4. Applications for quantum sensing. Left panel: Expectation values of the phase-shift measurement when the GHZ state of $N \equiv 8$ transmons is obtained with robust (solid line) and non-robust (dahsed line) optimal control . The line for robust optimal control is much closer to the ideal $\cos(N\theta)$ behaviour. Right panel: Variance of the measured phase shift achieved with robust (solid line) and non-robust (dashed line) optimal control. Robust optimal control helps reduce the error greatly and keep it very close to the ideal $1/N^2$ Heisenberg bound.

optimal control leads to an expectation value close to the ideal $\cos(N\theta)$ behaviour, non-robust optimal control gives more visible discrepancy due to the significant deviation of the actual state and the GHZ state. The precision (variance) obtained with robust optimal control reproduces closely the Heisenberg limit, $1/N^2$, for small $\theta$. Non-robust optimal control, on the other hand, results in a much larger error, more than five times larger for small $\theta$. This shows that quantum sensing can benefit greatly from utilising robust optimal control for generating the multi-qubit entangled states. This is particularly important when the number of qubits is increased for achieving a higher precision, resulting in a larger number of uncertain parameters.

In practical situations the pulse duration is limited by the coherence time, $T_2$, of the qubits. It is known that the collective decoherence rate of the GHZ state is enhanced by a factor of $N$ where $N$ is the number of qubits [15], giving rise to a lifetime of $T_2' = T_2/N$. The upper limit in the fidelity can be approximated by $F \leq 1 - T/T_2' \equiv 1 - NT/T_2$. Therefore, to realise the GHZ state with a fidelity $F$ the pulse duration needs to be shorter than $T_2(1 - F)/N$. The typical coherence time of transmons is around 100 $\mu$s. For the star graph of 8 transmons in our calculation $F = 0.92$ and hence the upper limit for the pulse duration is 1 $\mu$s which is well above the 400 ns used in our calculation. Thus, the effect of decoherence is negligible. We also note here that the precision enhancement from using entangled states exists only in ultra-fast sensing situations where the sensing time must be much smaller than the coherence time $T_2$ of a single qubit [15].

## V. CONCLUSIONS

To conclude, we develop an algorithm for the robust optimal control of an interacting quantum many-body systems of 10 qubits and 9 uncertain parameters. We demonstrate that a GHZ state of 10 two-level qubits (8 multi-level transmons) can be realised with over 99.9% (90%) fidelity despite 5% uncertainty in all qubit-qubit interaction strengths. In both cases robust optimal control improves the fidelity of the state preparation process by more than one nine compared with non-robust optimal control. When this GHZ state is used for quantum sensing, we show that robust optimal control greatly improves the measurement precision and is crucial for achieving the Heisenberg bound. The exact Krylov-subspace method for computing the unitary evolution in our optimisation can be scaled up to 20 qubits. Extending to larger systems requires approximate methods for simulating the quantum dynamics such as tensor networks [16, 17] or neural networks from machine learning [18]. We demonstrate how the exponential complexity in the number of uncertain parameters can be avoided by utilising the symmetry of the target state and the multi-qubit system. For the case of a GHZ state on a star graph this complexity is only linear.

### ACKNOWLEDGMENTS

This work is supported by the UK Hub in Quantum Computing and Simulation, part of the UK National Quantum Technologies Programme with funding from UKRI EPSRC grant EP/T001062/1, and the EPSRC strategic equipment grant no. EP/L02263X/1. All the codes and simulation data of this paper are available without restrictions [19].

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
