# Peer review of "Robust optimal control of interacting multi-qubit systems for quantum sensing"

_SciPost Physics_

## Round 1 · Referee Report · Anonymous (Referee 1) · 2022-3-24

Strengths

  1. The paper is clear in formulating the problem of robust quantum control.
  2. The problem, in general, is highly relevant, and the "sequential convex programming" approach used to solve it is quite interesting.
  3. The paper provides all the necessary details and parameters to reproduce the results

Weaknesses

  1. The authors make some very strong assumptions that do not seem warranted in general, limiting the applicability of the approach. A general discussion of when the assumptions hold or break down for superconducting circuits is missing.
  2. It is not entirely clear what aspects of the work are novel. The fundamental method has been described in the author's previous publications and the example might seem contrived without further motivation

Report

The authors develop a robust optimal control approach to create a GHZ state that provides Heisenberg limit (HL) precision for quantum sensing. The authors study this for a star graph of capacitively coupled transmons. There are two central claims in the paper.

  1. Robust optimal control can be achieved efficiently via sequential convex programming by evaluating the vertices of the uncertainty region. Furthermore, the effective number of vertices is often less than exponential due to symmetries in the system.

  2. Applying this to the generation of a GHZ state, the method leads to robust metrological Heisenberg scaling.

The first point depends on the observation that the fidelity is convex in a small enough region (a hypercube) around the perfect physical parameters of the system. Then, the worst-case fidelity parameters lie on the boundary (hypersurface), reducing by one dimension in the region where the robustness has to be improved. The authors checked this in the specific system they considered (a star graph of transmons) using a 5% (that is, ±2.5%) deviation in the qubit-qubit coupling. In general, I agree that convexity is expected for a sufficiently small region around the optimum. However, it is unclear how that region extends. Certainly, there is a limit to the convexity. I have tried to reproduce the simulations shown in the manuscript for a simplified system of three transmons truncated to three levels in the "star" configuration (one at the center, and one on the left/right, in this case). I see a convex landscape, starting from a control field obtained by optimizing for a GHZ state in the ideal system and simulating the dynamics for 100×100 points of variations in the coupling with a variation of ±2.5%. Still, this convexity degrades for ±5%. I'm attaching a figure showing this. In general, the amount of variation is determined by the circuit engineering quality, and it is not something that can just be chosen arbitrarily. I expect that adding additional transmons to the system and especially allowing variation in other system parameters besides the qubit-qubit interaction would make the optimization landscape potentially far less convex.

This makes the method far less general than the authors present it as. They could remedy this by analyzing in some detail for typical superconducting circuits what actual realistic variations in system parameters are, based on the state of the literature, and why for that realistic range of parameters a convex landscape might be expected.

Assuming the relevant region of the parameter landscape is indeed convex, the authors also claim that it is sufficient to consider only the vertices of the hypercube (minimum and maximum value of each parameter). While I agree that convexity reduces the relevant points from a hyper-volume to a hyper-surface, it is not clear why considering only the vertices of that surface is sufficient. Is it always true that the worst fidelity is in a vertex? Is this only an approximation? Does the optimization preserve this?

Regarding the second point, the robustness of the metrological gain for the transmon start system, the sensitivity looks robust indeed. However, the right panel in FIG. 4 shows sensitivity beyond the HL at $\theta/\pi = 0.1$ ($N^2 \Delta \theta^2 < 1$). That cannot be as it surpasses the lowest bound. There must be an error in the plot or the code.

Lastly, the manuscript is not entirely clear on what novel results it presents. Fundamentally, the idea of sequential convex programming was developed in earlier publications by the same authors. Those previous publications are a bit vague on how the convex hypercube it is being sampled, so it might be that the novel aspect in this manuscript is that sampling only the vertices of the hypercube is sufficient. If this is the case (and assuming the authors can dispel my skepticism that this is, in fact, true), they should clearly state this in the abstract/introduction/conclusion.

Alternatively, the paper could also focus on the specific application of generating a GHZ state in the star-transmon system for the purpose of quantum metrology. In this case, that manuscript would have to explain in more detail why that system is of practical interest, as opposed to an example specifically chosen because the optimization methods perform well for that specific configuration and those specific parameters. Also, while the GHZ state in the ideal case does provide Heisenberg scaling, the GHZ is a notoriously unstable state for quantum metrology. In a system with multiple transmons, as described in the manuscript, dissipation would probably not be negligible and would break the GHZ state.

I will enumerate further points that the authors should address in the changes section.

In general, I would only be able to recommend publication after a major revision that fully addresses my fundamental concerns about the validity of the results.

Requested changes

  1. An exact approach exists to calculate the gradients (Eq. (5-7)) consisting in, see Van Loan, IEEE Trans Automat. Control 23, 395 (1978) and Floether NJP 14, 073023 (2012). If that approach was used, the phrase "exact gradients" on page 3 (line 198) would be justified, but when using Eqs (5-7), the word "exact" is best avoided.
  2. Sec. II b is a bit confusing. Is it c a random initial guess or the result of an optimization? The paragraph could use some rewriting.
  3. The authors state that it is important to consider enough levels in the time evolution. How many in this case? Please specify how many transmon levels were included in the simulation.
  4. The authors argue that reaching the GHZ state with high precision is crucial for achieving HL precision. They show that a coherent state gives SQL scaling to justify that claim. That does not support the argument at all. A deviation in the system's parameters will lead to a final state around the target, but the overlap between the GHZ and the coherent state is $(1/2)^{N-2}$ which becomes very small very quickly. Perhaps a linear combination between the two would be a better example to make that point or FIG. 4 itself.
  5. It would be nice to have the metrological gain in dB at the conclusion for the best and worst cases.

Typos:

  • Schrodinger in line 16 should be Schrödinger
  • "qubits" in line 25 should be "qubit"
  • drop "the" in line 172
  • "fist" should be "first" in line 179

Attachment

---

## Round 1 · Referee Report · Anonymous (Referee 2) · 2022-3-25

Report

In their manuscript the authors study a system of several transmons (qubits) in a star configuration. The identical qubits are coupled to the central qubit via capacitive coupling modelled as flip-flop interaction. Robust optimal control pulses allow to create a GHZ state on this system with robustness against the offset noise in the coupling constants. While the topic is interesting and timely and technically the work seems to be sound for the most part (see comments) there is not much novelty and it is not immediately clear whether the results are applicable beyond the numerical study of the transmon star graph. My major objections are:

The literature and state-of-the-art is not well presented.
The authors only cite 19 references despite the fact that there is a tremendous amount of work on optimal control, on transmon qubits and on quantum sensing and a connection to such earlier work should be made.

The method is not new.
Robust optimal control is a known method and has been applied, e.g., by Timoney et al. [Phys. Rev. A 77, 052334 (2008)] or in a very similar way as in the author’s manuscript by Said & Twamley [Phys. Rev. A 80, 032303 (2009)] to name just two works from more than a decade ago, but also their own Ref. 9 and Ref. 23 therein. This is of course no problem in principle but colliding with a central claim in the abstract. The authors should reformulate and – if possible – argue where their method is different and why it is an improvement.

The experimental realization might be difficult.
While the resulting fidelities are impressive and robust against uncertainties in the capacitive coupling between the qubits, it is not clear to which extent these high fidelities could survive in an experiment, where the transmons are not identical and also the level transition frequencies are subject to uncertainties (the reference Song et al. reports these uncertainties). Furthermore, the control pulse might be distorted due to the transfer function of the control apparatus. Wittler et al. discuss the difficulties to overcome these imperfections for superconducting qubits [Phys. Rev. Applied 15, 034080 (2021)]. This should be addressed.

The application is not realistic or not sufficiently discussed.
The authors claim that their GHZ state could be used for sensing at the Heisenberg limit. Only at the very end of Section IV. they mention that the enhancement due to entanglement exists only in ultra-fast sensing situations, but it is not clear whether or when such ultra-fast sensing is necessary. It should be discussed more explicitly.
Referring to the results presented in Fig. 4 the authors claim that the optimized state preparation brings a more than five times larger precision for small theta. However, it is not clear why one would operate the sensor in this disadvantageous parameter regime. Indeed, as discussed in the reference Degen et al., one can often optimally choose the parameter regime where the sensor offers the best sensitivity: in this case, e.g., by changing the sensing time or by applying an offset field. The authors should discuss this.

Minor comments:
In Eqs. 11-13 and Table 1 it is not clear to me if all indices and superscripts are correct and correctly introduced. For example, what is \omega_j^{(01)} and (though almost obvious) how are the variables with index related to the ones without index in the table?

Above Eq. (4) a claim on concavity is made and at this point it is not clear that later on it is checked only numerically or what happens when the uncertainty is different from 5%.

---

## Editorial Decision

awaiting_resubmission